# Suppression by RNA Polymerase I Inhibitors Varies Greatly Between Distinct RNA Polymerase I Transcribed Genes in Malaria Parasites

**DOI:** 10.3390/pathogens13110924

**Published:** 2024-10-24

**Authors:** Hermela Samuel, Riward Campelo Morillo, Björn F. C. Kafsack

**Affiliations:** 1Department of Microbiology & Immunology, Weill Cornell Medicine, New York, NY 10021, USA; 2ACCESS Summer Internship Program, Weill Cornell Medicine, New York, NY 10021, USA; 3Carleton College, Northfield, MN 55057, USA

**Keywords:** RNA Polymerase I, transcription, inhibitors, non-coding RNA, ribosomal RNA, malaria parasites, *Plasmodium falciparum*

## Abstract

The transcription of ribosomal RNA (rRNA) by RNA Polymerase I (Pol I) is the rate-limiting step in ribosome biogenesis and a major determinant of cellular growth rates. Unlike other eukaryotes, which express identical rRNA from large tandem arrays of dozens to hundreds of identical rRNA genes in every cell, the genome of the human malaria parasite *Plasmodium falciparum* contains only a handful single-copy 47S rRNA loci that differ substantially from one another in length, sequence, and expression in different cell types. We found that the growth of the malaria parasite was acutely sensitive to the Pol I inhibitors 9-hydroxyellipticine and BMH-21 and demonstrated that they greatly reduce the transcription of 47S rRNAs as well as the transcription of other non-coding RNA genes. This makes P. falciparum only the second known organism where RNA Polymerase I transcribes genes other than the 47S rRNAs. We found that the various types of Pol I-transcribed genes differed by more than two orders of magnitude in their susceptibility to these inhibitors and explored the implications of these findings for the regulation of rRNA in *P. falciparum*.

## 1. Introduction

Ribosomes are macromolecular machines that translate the genetic information within messenger RNA into proteins that carry out the vast majority of biological functions [1,2,3]. Given the ribosome’s central role in cellular function, eukaryotic genomes encode hundreds of identical ribosomal RNA (rRNA) genes that are organized into large tandem arrays, and the ribosome composition tends to be highly uniform across cell types [4,5,6].

Malaria parasites represent a striking exception to this rule. Despite having similarly sized genomes as budding yeast, which has a single array of 150 identical rRNA copies [7], the genomes of malaria parasites generally contain four to six rRNA loci, each located on a different chromosome and encoding only a single 47S rRNA gene [8,9,10,11]. Moreover, this small number of genes encode two to three distinct forms of 47S rRNAs that differ substantially from one another in length, sequence, and expression throughout the parasite lifecycle. The major human malaria parasite *Plasmodium falciparum* has five complete 47S rRNA genes. Two nearly identical A-type rRNA loci (A1 and A2 on chromosomes 5 and 7) are expressed in the liver and asexual blood stages, the S1 locus on chromosome 1 that is primarily expressed during gametocyte development and two nearly identical S2-type loci (S2a and S2b on chromosomes 11 and 13) that are highly expressed in the mosquito stages [11,12,13]. Regulation of the S2 loci is also quite distinct in *P. falciparum*. Under standard culture at 37 °C, these loci are only weakly expressed but their expression increases rapidly if parasites are exposed to the same temperatures experienced in the mosquito [14,15,16].

In eukaryotes, transcription is carried out by three RNA polymerase complexes (Pol I, II, and III) that share some subunits but have distinct catalytic cores, promoter types, and recruitment machinery [5,17]. Protein coding genes are transcribed by Pol II to generate capped and polyadenylated messenger RNAs that are targeted to the ribosomes for translation. Pol III transcribes the 5S rRNA, transfer RNAs, and other small non-coding RNAs with a host of functions. In contrast, Pol I only generates a single transcript in virtually all eukaryotes: the 47S rRNA that is processed into the 28S, 18S, and 5.8S rRNA that form the catalytic core of the ribosome. Despite this specialization, transcription by Pol I comprises up to 60% of the total transcriptional activity of growing cells [18]. The transcription of the 47S rRNA is the rate-limiting step in ribosome biogenesis [19], a major determinant of cellular growth rates [20,21,22], and commonly upregulated in a variety of cancer types. Unlike the activity of the other two RNA polymerases, transcription by Pol I is insensitive to inhibition by α-amanitin [23] but Pol I-specific inhibitors have recently been developed for use in oncotherapy [24]. To date, the African trypanosome *Trypanosoma brucei* is the only known organism where Pol I transcribes genes other than the 47S rRNA. In these organisms, Pol I also generates polycistronic transcripts from specialized expression site loci, which encode the proteins that form the parasite’s dense surface coat [25].

Since malaria parasites are highly unique in their diversity, organization, and expression of the 47S ribosomal RNA and in the absence of validated RNA Polymerase I inhibitors, we tested the effects of these inhibitors on growth and Pol I transcription in the most widespread and deadly human malaria parasite, *P. falciparum*.

## 2. Materials and Methods

### 2.1. Parasite Culture

In this study, we used the *Plasmodium falciparum* parasite strain NF54 (MRA-1000) obtained from BEI Resources. Parasites were maintained following established cultured conditions [26] using 0.5% AlbuMAX II (Fisher Scientific, Carlbad, CA, USA, Cat #50-114-7584) supplemented malaria complete media and kept at 37 °C under 5% O_2_, 5% CO_2_, and 90% N_2_.

### 2.2. P. falciparum Growth Inhibition Assay

A fluorescence-based parasite growth-inhibition assay was carried out according to previously described protocols with minor modifications [27,28]. Briefly, ring-stage parasite cultures were adjusted to 0.5% parasitemia and 2% hematocrit before being seeded into a flat-bottom 96-well plate in the presence of increasing drug concentrations at a final volume of 200 µL of malaria complete media. Parasites were allowed to grow at 37 °C for 72 h. Upon incubation time, 150 µL from each well was transferred to a black flat-bottom plate, and 100 µL of SYBR Green lysis buffer (20 mM Tris, pH 7.5; 5 mM EDTA, 0.008% *w*/*v* saponin, 0.08% *v*/*v* Triton X-100, and 1X SYBR Green [Invitrogen, Carlbad, CA, USA, Cat #S7563]) was added to each well. The plate was then incubated with gentle agitation for 1 h, protected from light. Fluorescence (excitation at 485 nm/emission at 538 nm) was measured using a Molecular Devices SpectraMax iD5 plate reader (San Jose, CA, USA). Fluorescence units (FU) from three technical replicates were normalized to chloroquine (50 nM) and DMSO-treated controls (0.5% final concentration), averaged, and plotted using GraphPad Prism 9 software. A four-parameter dose–response curve was fitted, and the EC50 was determined. Reported EC50 values were averaged from three independent experiments, with standard error of mean values.

### 2.3. RNA Polymerase I Inhibition Assay 

Cultures of trophozoites synchronized to 22 ± 3 h-post-invasion were adjusted to 1.5% parasitemia, equally split, and treated with increasing concentrations of the RNA Pol I inhibitors 9-HE (5,11-dimethyl-6H-pyrido[4,3-b]carbazol-9-ol, monohydrochloride) CAS 52238-35-4 (Cayman Chemical, Ann Arbor, MI, USA) and BMH-21 (N-[2-(dimethylamino)ethyl]-12-oxo-12H-benzo[g]pyrido[2,1-b]quinazoline-4-carboxamide) CAS 896705-16-1 (Cayman Chemical, Ann Arbor, MI, USA). Both treated and untreated parasites (with the same volume of DMSO added) were incubated for 3 h at 26 °C. In parallel, untreated control parasites were kept at 37 °C. After the incubation period, the parasites were harvested and washed once with 1xPBS before proceeding to RNA isolation.

### 2.4. RNA Isolation, cDNA Synthesis, and qPCR

Infected erythrocyte cultures were spun down (3 min at 800 g) and resuspended in PBS containing 0.05% saponin to lyse the host cells. After 5 min at room temperature, released parasites were spun down (3 min at 1500 g) and washed once with PBS. Total RNA from saponin-lysed parasites was extracted by the TRIzol reagent (Invitrogen, Carlbad, CA, USA, Cat #15596026) and using the PureLink RNA Mini Kit (Invitrogen, Carlbad, CA, USA, Cat #12183018A) following the manufacturer’s instructions. cDNA was synthesized from 500 ng of total RNA pretreated with DNAse I (amplification grade, ThermoScientific Invitrogen, Carlbad, CA, USA, Cat #EN0521) using SuperScript III reverse transcriptase (Invitrogen) and random hexamers. Quantitative PCR was performed on the Quant Studio 6 Flex (Invitrogen Invitrogen, Carlbad, CA, USA, Cat #18068015) using iTaq SYBR Green (BioRad, Hercules, CA, USA, Cat #1725120) with specific primers for selected targets (Appendix A) and normalized to seryl-tRNA synthetase (PF3D7_0717700). qPCR Primers were selected from earlier studies for *48S rRNAs* [15], *truRNA* [16], uce (PF3D7_0812600) [29], and *seryl-tRNA synthetase* [30] or newly designed using Primer 3 [31] to yield products sized 150–300 nt with annealing temperatures matching the other primer sets. The target specificity and efficiency of the newly designed primer sets were validated using genomic DNA and melting curve analysis.

## 3. Results and Discussion

### 3.1. Inhibition of P. falciparum RNA Polymerase I Activity by 9-HE and BMH-21

We decided to test the effect of 9-hydroxyellipticine (9-HE) and BMH-21, two compounds that inhibit RNA Pol I in mammalian cells [32,33,34]. 9-HE was previously shown to inhibit the growth of *Plasmodium falciparum* [35] and *Trypanosoma cruzi* [36]. BMH-21 was shown to inhibit growth and RNA Pol I transcription in African trypanosomes [34] but its anti-malarial activity has not been reported. We found that both compounds were able to inhibit the intraerythrocytic replication of *P. falciparum* at sub-micromolar concentrations with EC50 values of 56 ± 3 nM for 9-HE and 352 ± 64 nM BMH-21 (Figure 1).

To test if these compounds can inhibit RNA Pol I transcription specifically, we monitored the transcription of 47S rRNAs, the RNA Pol III-transcribed 5S rRNA, and mRNA transcribed by RNA Pol II (Figure 2). Under standard conditions, only the nearly identical A1 and A2 loci are actively transcribed in asexual blood stages while the S1 and S2a/b loci are expressed in gametocytes and mosquito stages [8]. However, transcription of the S2 loci can be induced in the asexual blood stage by simply shifting parasite cultures to the temperature experienced in the mosquito host, thus allowing us to monitor new transcription by RNA Pol I [14,15,16]. To monitor the nascent transcription of 47S rRNA, we used primers that bind within the 5′ external transcribed spacer (5′ ETS) that sits upstream of the 18S sequence and is rapidly cleaved and degraded during ribosomal biogenesis [37,38].

Consistent with these earlier reports [15,16], we found that shifting parasite cultures to 26 °C for 3 h greatly induced the expression of S2 47S rRNA and the long non-coding *truRNA* relative to their expression at 37 °C (Figure 2A). The transcription of A1 and A2 47S rRNA decreased 2- to 4-fold while the transcription of S1 47S rRNA remained largely unchanged. The temperature shift had only minor effects on the 5S and the housekeeping ubiquitin-conjugating enzyme (*uce*, PF3D7_0812600) genes that are transcribed by RNA Pol II and Pol III, respectively. Concurrent treatment with 12.8 µM 9-HE or BMH-21 substantially reduced the expression of RNA Pol I transcripts but had no significant effect on the representative Pol II and Pol III transcripts, indicating that both compounds specifically inhibit transcription by RNA Pol I in *P. falciparum* (Figure 2B).

### 3.2. Susceptibility to Inhibitors of RNA Pol I Activity Differs Greatly Between Pol I Transcribed Genes

Interestingly, we found that the transcription of S1 rRNA was less affected by 9-HE and BMH-21 relative to the other four Pol I transcripts (Figure 2B). To test if these other transcripts might also be differentially susceptible to inhibition by either inhibitor, we measured their effect on Pol I-transcribed genes across a wider range of concentrations. Indeed, treatment with increasing concentrations of 9-HE or BMH-21 revealed substantially different responses between these RNA Pol I-transcribed genes (Figure 3). The transcription of A-type 47S rRNAs was acutely sensitive with half-maximal inhibition concentrations in the 30 nM range for 9-HE and 140 nM for BMH-21. These concentrations are a close match to those inhibiting growth, strongly suggesting that the mechanism of action of 9-HE and BMH-21 is indeed the inhibition of A-type rRNA transcription, which form the large majority of ribosomes in asexual blood stages. Strikingly, inhibiting the transcription of the other genes required much higher concentrations of 9-HE with half-maximal concentrations of 290 nM, 720 nM, and 4.9 µM for *truRNA*, S2 rRNA, and S1 rRNA, respectively. The inhibition of transcription by BMH-21 at these loci followed the same pattern (A > S2 = *truRNA* > S1) but was approximately 4-fold less susceptible compared to 9-HE. The fact that *truRNA* transcription could be inhibited by two Pol I inhibitors makes *Plasmodium falciparum* only the second organism in which Pol I has been demonstrated to produce transcripts other than the 47S ribosomal RNA.

Intriguingly, the transcription of S1 and S2 rRNAs increased at lower concentrations that effectively inhibited transcription of A-type rRNAs but not transcription of the *truRNA*. We, therefore, observed four distinct patterns of susceptibility to these Pol I inhibitors: (i) high susceptibility for A-type rRNA, (ii) intermediate susceptibility with enhanced transcription at concentrations that already inhibit A-type rRNA for S2 rRNA, (iii) intermediate susceptibility without enhancement for the *truRNA*, and (iv) low susceptibility with an enhancement for S1 rRNA.

The observation that the transcription of these genes differs substantially in their susceptibility to Pol I inhibitors suggests a sequence dependence of this activity. Both inhibitors have been shown to intercalate into GC-rich DNA [32,39] but with different effects. Treatment with BMH-21 most strongly alters transcription elongation by Pol I and leads to the stalling of Pol I upstream of G-rich stretches in the template [32], while 9-HE specifically altered the promoter binding of the SL1 promoter recognition complex required for the recruitment of Pol I and transcription initiation [33]. Fang and colleagues noted that the transcription start site of A1 and A2 rRNA genes is immediately preceded by a 20nt stretch of 75% GC content that is absent from the S1 and S2a/b promoters [15], providing a likely explanation for the greater inhibitor sensitivity of A-type transcription we observed. Interestingly, the 100bp immediately upstream of the S2a and S2b transcription start sites contains six regularly spaced deoxyguanosine di-nucleotides (14, 16, 16, 17, and 19 bp apart) that stand out within this region of only 18% GC content, while no such evenly spaced repeats could be found close to the *truRNA* or S1 rRNA transcription start sites.

The observed enhancement of transcription from S-type promoters at concentrations that inhibited transcription from A-type promoters suggests that one or more of the components of the transcriptional machinery is available in limited supply. Reduced transcription at the more susceptible A-type loci then frees these factors up to drive maximal transcription of the S-type promoter. Intriguingly, the lack of enhancement at the same concentrations would imply that this shared component is not limiting for maximal transcription of the *truRNA* promoters.

### 3.3. lncRNAs Downstream of the truRNA Likely Derive from a Single Longer Transcript

The current *P. falciparum* genome annotations (PlasmoDB release 68) include three non-coding RNA genes (PF3D7_1148100/200/300, referred to as regions *a*/*b*/*c* hereafter) immediately upstream of the S2b locus and three additional non-coding RNA genes (PF3D7_1370500/600/700, referred to as regions *d*/*e*/*f* hereafter) upstream of the S2a locus. Of these, regions *b*–*f* lay upstream of the S2a and S2b rRNA genes on chromosomes 13 and 11 within 15.4 kb regions that share 98.6% sequence identity, while region *a* falls outside this region of homology and only presents on chromosome 11 (Figure 4A). Regions *b* and *d* are contained within the gene that produces the recently described 2.8 kb *truRNA* [16]; however, the current genome annotations do not yet account for those findings. Region *c* lies downstream of the *truRNA* and is homologous to sequences within region *e.*

Since both the S2-type rRNA and the *truRNA* are rapidly upregulated by exposure to lower temperatures, we wanted to test whether the transcripts from regions *a*, *b*/*d*, *c*/*e*, or *f* might be similarly regulated. We detected substantial transcripts being made from all regions (Figure 4B) but found that only the regions overlapping with or downstream of the *truRNA* were cold-inducible while the expression of PF3D7_1148100 was effectively unchanged (Figure 4C). The transcription of PF3D7_1148100 in region *a* was also not sensitive to RNA Pol I inhibitors, indicating that it forms a distinct transcriptional unit transcribed by either RNA Pol II or III. The inhibition patterns for regions *b*–*f* closely mirrored that of the *truRNA*. While regions *c*/*e* and *f* had distinct steady-state levels from the *truRNA* (Figure 4B), the fact that they exhibited highly similar upregulation in response to the temperature shift and their susceptibility to RNA Pol I inhibitors led us to wonder whether they might be transcribed from a single promoter prior to subsequent processing into fragments with variable stability. Indeed, we were able to amplify transcripts bridging from the *truRNA* to region *c*/*e* and from region *c*/*e* to region *f* but not from the *truRNA* to region *f* or from region *f* into the *S2a rRNA* (Figure 4E), supporting a single, rapidly processed transcript extending from the *truRNA* into region *f.* The *truRNA* itself was already shown to be processed from a 2.8 kb to a mature 1.3 kb [16]. We plan to use long-read RNA sequencing to map the full-length transcript in future studies.

When combined with the inability to amplify a product bridging region *f* and the S2 transcript, the distinct inhibitor susceptibility patterns of the upstream transcripts and the S2 strongly support their transcription from distinct promoters (Figure 4D). Current annotations also include a non-coding RNA gene (PF3D7_0531500) immediately upstream of the region encoding the A2 18S rRNA on chromosome 5. However, this annotation should be removed as it is downstream of the A2 rRNA transcription start site and thus part of the A2 5′ ETS [15].

### 3.4. P. falciparum Lacks Identifiable Homologs to Many RNA Pol I-Specific Components

Our findings show that the activity of the distinct RNA Pol I promoters in malaria parasites differs not only in response to temperature but also in their sensitivity to RNA Pol I inhibitors. We therefore tried to identify orthologs to components of Pol I recruitment and transcription machinery in mammals, budding yeast, and the protozoan *Trypanosoma brucei*, where this process has been well-studied [5,40,41,42,43,44]. A search for these components in the *P. falciparum* genome only found orthologs to RPA1 and RPA2, the two subunits that form the catalytic core (Figure 5). While orthologs to the Pol I subunits shared with Pol II and Pol III were all present, none of the Pol I-specific subunits could be identified. Promoter recognition by Pol I depends on distinct complexes in the other systems (the SL1 complex in humans, the CF complex in budding yeast, and the CITFA complex in *T. brucei*) but, again, none of the Pol I-specific components appear to be present in the malaria parasites, including RBP5z and RBP6z, the RNA Pol I-specific paralogs of RBP5 and RBP6 found in *T. brucei* [41]. Pol I transcriptional machinery in malaria parasites is therefore either highly divergent or substantially reduced and warrants further investigation.

## 4. Conclusions

Unlike those of most eukaryotes, the genomes of malaria parasites contain multiple 47S rRNA genes that differ in length and sequence and are expressed in specific portions of the parasite lifecycle. In this study, we demonstrated that RNA Polymerase I of *P. falciparum* is susceptible to inhibition by 9-HE and BMH-21 but that the transcription of A-type 47S rRNA was one to two orders of magnitude more sensitive than the transcription of the S2-type or the S1 rRNAs. While the mechanisms underlying these differences warrant further investigation, we noted that both inhibitors are known to intercalate into GC-rich DNA and speculated that the observed differences in inhibition may be explained by the presence of a GC-rich region preceding the transcription start site A-type 47S rRNA that is absent from the S1- and S2-type rRNA promoters. While very little is known about how RNA Pol I is recruited to promoters in *P. falciparum*, this observation suggests that parasites may rely on distinct protein complexes for Pol I recruitment for the different 48S rRNA genes, making this a critical area of investigation for understanding the mechanisms underlying the expression of distinct rRNA throughout the lifecycle. Finally, we found that Pol I also transcribes non-coding RNA genes located upstream of the S2 rRNA genes in a temperature-responsive manner, making *P. falciparum* only the second known example where Pol I transcribes genes other than 47S rRNA.

## Figures and Tables

**Figure 1 pathogens-13-00924-f001:**
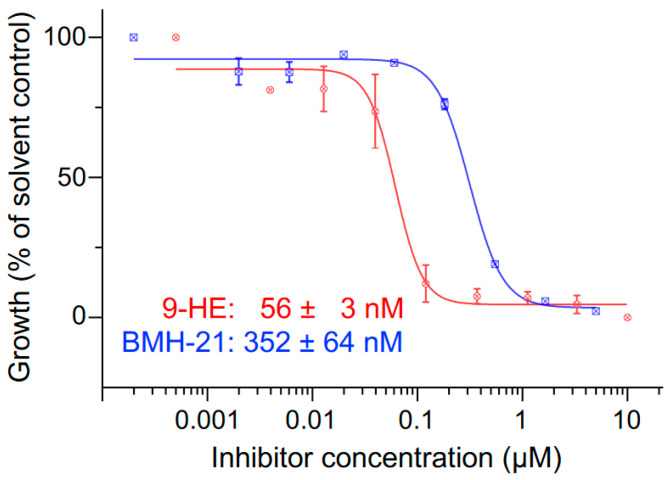
Growth inhibition of *P. falciparum* asexual blood stages by 9-HE (red) and BMH-21 (blue). The half maximal effective concentration (EC50) ± SEM was calculated from three independent biological replicates of 72-h SYBR Green Growth assays. The inhibition curves shown are representative replicates with data points indicating the mean and SEM from three technical replicates.

**Figure 2 pathogens-13-00924-f002:**
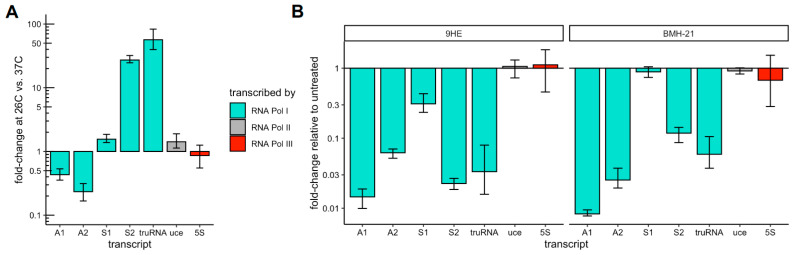
Validation of 9-HE and BMH-21 as specific inhibitors of RNA Polymerase activity in *P. falciparum*. (**A**) Growth at 26 °C for 3 h reduced transcription of A1 and A2 47S rRNA and greatly induced transcription of S2 47S rRNA and the long non-coding *truRNA* but had negligible effects on S1 47S rRNA, *uce* mRNA, and 5S rRNA transcription. Expression changes in RNA Pol I (turquoise), Pol II (grey), and Pol III (red) transcription after 3 h shift to 26 °C relative to expression at 37 °C were measured by qRT-PCR. To measure new transcription rather than steady-state levels, primers were targeted to the 5′ ETS of the 47S rRNAs, which are rapidly cleaved and degraded. (**B**) Concurrent treatment of samples shown in a) with 12.8 µM 9-HE (left panel) or BMH-21 (right panel) inhibits transcription of RNA Pol I transcripts (turquois) but not RNA Pol II (grey) or RNA Pol III (red). The means and non-parametric 95% confidence intervals based on three biological replicates are shown throughout.

**Figure 3 pathogens-13-00924-f003:**
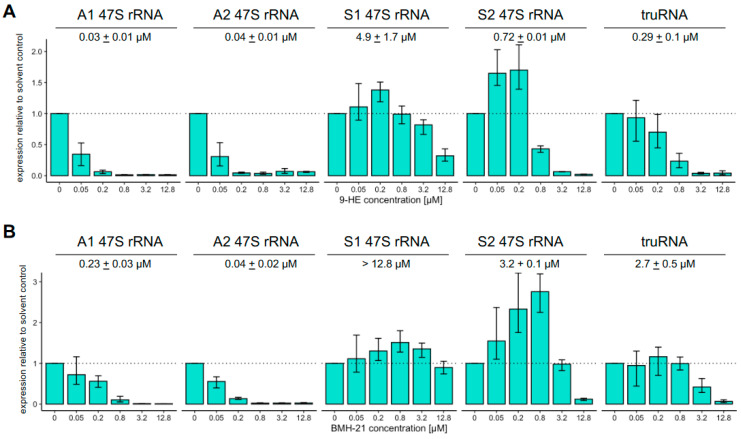
Inhibitor susceptibility of RNA Pol I transcription differs substantially between loci. Transcription of A1, A2, S1, and S2 47S rRNA or *truRNA* in the presence of five different concentrations of 9-HE (**A**) or BMH-21 (**B**) was measured by qRT-PCR and compared to solvent control (dotted line). Bars indicate the mean change relative to solvent control, the error bars indicate the non-parametric 95% confidence interval, and the numerical values shown are the mean half-maximal concentrations (IC50) based on three independent biological replicates.

**Figure 4 pathogens-13-00924-f004:**
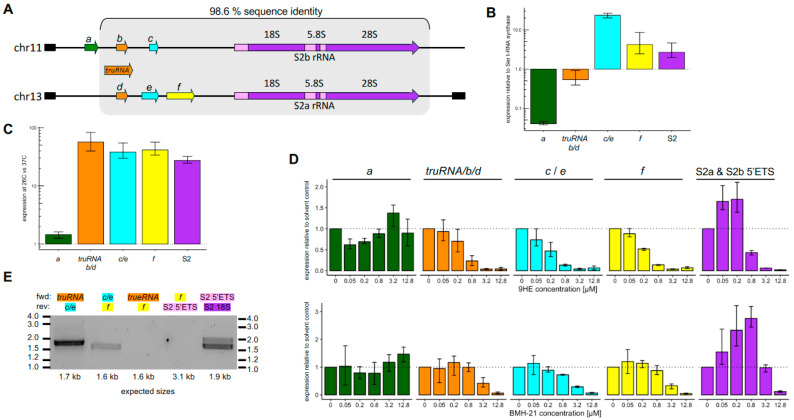
Regulation of S2-locus associated lncRNAs. (**A**) Organization of the S2a and S2b rRNA loci (purple) on chromosomes 13 and 11 along with annotated upstream non-coding RNAs. Six regions (*a*/*b*/*c* on chr. 11 and *d*/*e*/*f* on chr. 13) are annotated as non-coding RNA genes PF3D7_1148100/200/500 and PF3D7_1370500/800/900, respectively, in the current version of PlasmoDB (release 68). Of these, regions b–f fall within a 15 kb span of very high homology (grey) while region *a* is only found on chromosome 11 immediately upstream. Regions *b* and *d* (orange) are contained within the gene that produces the recently described 2.8 kb *truRNA*. The nearest protein-coding genes are indicated in black. (**B**) Transcript levels relative to seryl-tRNA-synthase. (**C**) Change in expression from regions *a*–*f* and S2 rRNA after a 3 h shift to 26 °C. (**D**) Transcription from regions *a*–*f* and S2 rRNA in the presence of five different concentrations of 9-HE (top) and BMH-21 (bottom) was measured by qRT-PCR and compared to solvent control (dotted line). (**E**) RT-PCR products from primers spanning the indicated regions amplifying cDNA from the solvent control sample. Expected sizes are indicated below each lane and primers targeting the nascent S2 47S rRNA were used as a positive control. Amplification from a “no RT” control yielded no products. All bars indicate the mean and the error bars show the non-parametric 95% confidence interval based on three independent biological replicates. Results in (**B**–**D**) are based on primer pairs BKO_2737/2738, BKO_2674/2675, BKO_2735/2736, BKO_2737/2738, and BKO_2672/2673. Lanes 1–5 in (**E**) were amplified using primer pairs BKO_2674/2736, BKO_2735/2738, BKO_2674/2738, BKO_2737/2673, and BKO_2672/2696.

**Figure 5 pathogens-13-00924-f005:**
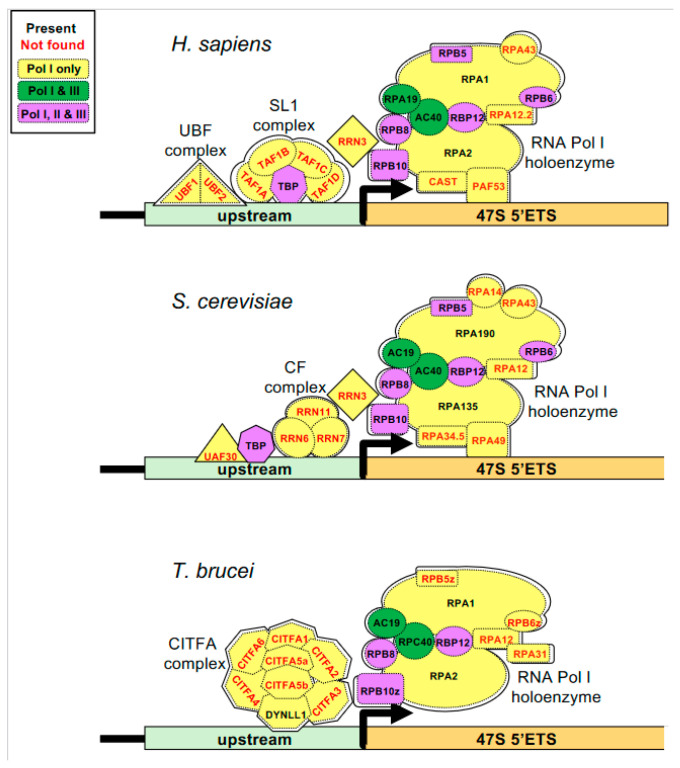
*P. falciparum* lacks identifiable homologs to many RNA Pol I-specific components. Subunits specific to RNA Polymerase I holoenzyme or the promoter recognition complexes are shown in yellow, those shared with RNA Pol III in green, and those shared by RNA Pol I, II, and III in purple. Black names indicate subunits with identifiable orthologs in *P. falciparum* while no clear ortholog exists for those with red names.

## Data Availability

All data are contained within the manuscript.

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
