# Peer review of "Suppression by RNA Polymerase I Inhibitors Varies Greatly Between Distinct RNA Polymerase I Transcribed Genes in Malaria Parasites"

_pathogens, 2024, doi:10.3390/pathogens13110924_

Round 1
Reviewer 1 Report
Comments and Suggestions for Authors
In this manuscript, the authors report that the growth of Plasmodium falciparum parasites is sensitive to two Pol I inhibitors: 9-hydroxyellipticine (9HE) and BMH-21. They show that these inhibitors not only affect the transcription of the 47S rRNA, but also the transcription of some other non-coding RNAs located upstream of the rRNA genes, that are also transcribed by Pol I. They demonstrate that these non-coding RNAs show different susceptibility to the inhibitors.
Major issue
Although the manuscript is interesting, in my opinion, it does not offer enough new and relevant information to deserve publication in Pathogens. It would be required that the authors include some extra results to strengthen the manuscript. In this regard, the authors should determine the transcriptional nature of the upstream non-coding RNAs in the S2a and S2b loci. The transcription start site(s) must be identified to determine if a single or several transcripts are produced from regions b-c (on chromosome 11) and from regions d-f (on chromosome 13). Is region “a” independently transcribed? It is confusing that the truRNA does not map exactly to the “b” and “d” regions. This issue should be clarified. As mentioned by the authors, long-read RNA sequencing could also help them to answer some of these questions.
Minor issues
Line 42. The chromosomal location of the S2a and S2b loci is inverted (according to Figure 4A).
Line 138. Primers that bind within the…
Line 159. Please indicate what protein is encoded by the uce gene.
Lines 195-197. The authors say that the observed differences in susceptibility to Pol I inhibitors could reflect differences in the Pol I complex composition that alters inhibitor susceptibility. Are there any reports about differences in the Pol I complex composition in any organisms?
Line 226. Do you mean “regions b and d”?
It is not clear why the experiment shown in Figure 4B was performed. I believe it is not mentioned in the text. Please clarify.
Please verify that the Figure 4 panels mentioned in the text (lines 246 to 245) are correct.
In section 4.4 (lines 270 to 279), please include references.
In Figure 5, the RPB5z and RPB6z subunits in T. brucei are Pol I-exclusive. Other isoforms, simply called RPB5 and RPB6, are restricted to Pol II and Pol III.
The quality of all figures should be improved.
Author Response
We have addressed all reviewer comments in the attached PDF.

Reviewer 2 Report
Comments and Suggestions for Authors
Article
Suppression by RNA Polymerase I Inhibitors Varies Greatly Between Distinct RNA Polymerase I Transcribed Genes in Malaria Parasites.
Authors present a very superficial study where the malaria parasite Plasmodium falciparum was treated with two RNA Polymerase inhibitors and resultant changes in different rRNAs was assessed. Article would be more compelling if authors made any attempt to outline why these results are of interest to the reader or scientific community at large.
Questions/Suggestions
Line 32: "budding yeast, which has a single array of 150 identical rRNA copies"
Citation(s) needed
Line 50: "In eukaryotes, transcription is carried out by three RNA polymerase complexes (Pol I, II, and III), that share some subunits but have distinct catalytic cores, promoter types and recruitment machinery."
Citation(s) needed
Line 66: "Since malaria parasites are highly unique in their diversity, organization, and expression of the 47S ribosomal RNA, we decided to explore the effect of these inhibitors on growth and Pol I transcription in the most wide-spread and deadly human malaria parasite P. falciparum."
Comment: "we decided to explore" is not great wording when giving reason(s) for doing a study. Studies should have a clear aim/hypotheses. Some definite fact to be determined from doing a study, and why knowing this fact is worth not only the researchers time to conduct said study, but also the readers time to read their article. This should also be clear in article abstract.
Line 72: "parasite strain NF54 obtained from BEI Resources"
please include cat# MRA-1000
Line 74: "AlbuMAX II (Gibco) supplemented malaria complete media"
please include cat#
Line 75: "kept at 37oC under 5% O2, 5% CO2, 90% N2."
please outline how this was achieved.
Line 81: " at a final volume of 200 μL."
Question: In what media?
Line 94: "Cultures of 22 hpi ± 3 hpi trophozoites"
Question: hpi to what?
Line 99: "were incubated for 3 hours at 26°C"
Question: Why 26°C treatment? Especially when "untreated control parasites were kept at 37°C"
Line 103: "3.4. RNA isolation, cDNA synthesis and qPCR"
This paragraph needs more depth, please fully outline all sequences determined, and better label the primers which were used for each sequence in the table (see Line 220 comment below)
Line 104: "saponin-lysed parasites"
Please outline this step
Line 104/106: "TRIzol reagent (Invitrogen) and using the PureLink RNA Mini Kit (Invitrogen)"; " DNAse I (amplification grade, ThermoScientific) using SuperScript III reverse transcriptase (Invitrogen) and random hexamers."
Please include catalogue numbers
Line 111: Table 1. Image very pixelated. please provide table as text file not image.
Line 112:
Comment: no stats used to analyze data?
Line 115: "We decided to test the effect of 9-hydroxyellipticine (9-HE) and BMH-21, two compounds that inhibit RNA Pol I in mammalian cells (Jacobs et al. 2022; Andrews et al. 2013; 116 Kerry et al. 2017)".
A little too vague for a scientific manuscript. Should be more like: "We first determined the ability of 9-HE and BMH-21 to inhibit RNA Pol I activity in our Plasmodium falciparum samples"
Line 117: "9-HE was previously shown to have activity against Plasmodium falciparum (Montoia et al. 2014) and Trypanosoma cruzi"
Question: What does "have activity" mean?
Line 119: "BMH-21 was shown to inhibit growth and RNA Pol I transcription in African trypanosomes (Kerry et al. 2017) but its anti-malarial activity has not been reported."
Comment: better
Lines 123/140/163/228/279, Figs1-5: Images very pixelated. please provide higher res images of figures.
Would also suggest reformatting items within some figures from a landscape to portrait arrangement (e.g. fig 2: from 2 rows with 5 columns to 4 rows with 3 columns)
Line 129: suggest change "To test if these compounds can inhibit RNA Pol I " to "To test if these compounds could inhibit RNA Pol I "
Comment: past-tense should always be used.
Line 129: suggest change "we monitored transcription of 47S rRNAs, the RNA Pol III-transcribed 5S rRNA, and mRNA transcribed by RNA Pol II. " to
"we monitored transcription of 47S rRNAs, the RNA Pol III-transcribed 5S rRNA, and mRNA transcribed by RNA Pol II (Figure 2). "
Line 129: whole paragraph
comment: honestly, not a fan of mixing results with discussion. Text is little hard to unravel authors results vs what is known from previous studies.
Line 145: " on "
remove
Line 220: ". lncRNAs downstream of the truRNA likely derive from a single longer transcript."
Comment: This section was hard to understand, methods section appeared lacking in depth to help.
e.g.
"The current P. falciparum genome annotations (PlasmoDB release 68) include three non-coding RNA genes (PF3D7_1148100/200/300, referred to as regions a/b/c hereafter)"
Question: what primers were used for a/b/c?
"upstream of the S2b locus and three additional non-coding RNA genes (PF3D7_1370500/600/700, referred to as regions d/e/f hereafter) "
Question: what primers were used for d/e/f?
Line 281:
Manuscript lacks any meaningful conclusion statement/paragraph. Why should anyone care about these results? What are possible future directions? Strengths/weaknesses of study?
Author Response

(The authors gave the same response as above.)

Reviewer 3 Report
Comments and Suggestions for Authors
This paper is a description of the affect of Pol I inhibitors on the malaria parasite. It may be of interest to a small number of investigators and is consistent with previous work on the expression of the 47S rRNA transcript.
Not clear what the authors are trying to say in the paragraph at lines 213-219. If the target of the inhibitors is GC-rich DNA (ie, the template), what are these hypothetical shared components that are being titrated by the drug? Very confusing and could probably be deleted.
lines 10-11, delete virtually every and change to eukaryotes. Considering the immense diversity of eukaryotes and the fact that virtually all work on eukaryotes has been done in yeasts and mammals, this may not be true.
line 17, delete Surprisingly
Likewise, it is not clear what the point of Figure 5 is and it seems beyond the scope of the paper. Yeasts and humans are within sister groups of the opisthokonts within the amorphean clade and are somewhat related to each other. Kinetoplastids are within a clade called the discobans and the apicomplexans are within the SAR clade. The relationship between the SAR, discobans, and amorpheans are unknown, and at the very base of the eukaryotes. Not sure what the authors are trying to compare here. It just looks like the accessory proteins of the Pol I complex are rather different in the various eukaryotic clades. Furthermore, what does this have to do with the inhibitors -- the focus of the paper?
minor comments:
line 49, a separate section is not needed
line 88, chloroquine is not capitalized
line 138, external spacer is not capitalized
line 145, delete second on
line 278, therefore
Author Response

(The authors gave the same response as above.)

Reviewer 4 Report
Comments and Suggestions for Authors
The study constitutes an investigation of the regulation of some growth related genes in Plasmodium falciparum in the presence of inhibitors. It is generally well structured and methodologically correct. Apart from some particular points mentioned below, I have a few general comments that I would expect to be considered bey the authors. My main suggestion focuses on the general view of the findings. A summarized conclusion is lacking for example. The authors should add some parts for the evaluation of their findings. How can they be used in future applications. This should be also added in the abstract. I generally recommend not to combine results and discussion, but since the author prefer to do so, an extended general conclusion is necessary
Specific comments
Although not very important, I do not really agree with dividing the Introduction in two parts. I suggest to merge them
Also, the scope paragraph should be enriched with a brief introduction the methodological approach. How was this effect studied? By gene expression analysis? Please add 2-3 lines
Lines 109-111. Were the primers designed in this study? If yes, how? Provide some details. If no add references. Also the amplicon sizes have to be added for each pair
Author Response

(The authors gave the same response as above.)

Round 2
Reviewer 1 Report
Comments and Suggestions for Authors
While the minor changes I requested were made by the authors, my major issue was not addressed. Therefore, I still believe that the paper does not offer enough new and important information to be published in Pathogens.
Author Response
Comment 1: While the minor changes I requested were made by the authors, my major issue was not addressed. Therefore, I still believe that the paper does not offer enough new and important information to be published in Pathogens.
Response: As structure of the non-coding RNAs upstream of the S2 rRNA loci (Fig. 4) is not the main focus of this article, the costs associated with the long-read RNA sequencing requested by the reviewer will require additional research funds and will have to be part of a future study. While we think the results in Figure 4 will be useful to other researcher interested in non-coding RNAs in malaria parasites, we could remove those results from the manuscript and leave the decision to the editor.
Our manuscript includes:
- First demonstration and validation of RNA Polymerase I inhibitors in malaria parasites.
- First demonstration that activity of these inhibitors differs based on promoter sequence context.
- Demonstration that in P. falciparum RNA polymerase I transcribes genes other than rRNA, making it only the second organism for which this has been shown.
- Identification of additional non-coding transcripts that are regulated by temperature.
While we share the reviewers high opinion of this journal, we feel strongly that the results described in our manuscript is a good fit for the journal's scope and aims and compare well those in other parasitology articles that were recently published in Pathogens (Impact Factor 3.3):
https://www.mdpi.com/2076-0817/12/12/1412
https://www.mdpi.com/2076-0817/13/7/541
https://www.mdpi.com/2076-0817/12/9/1113
https://www.mdpi.com/2076-0817/12/6/803